# Atypical Mycobacteriosis Due to *Mycobacterium abscessus* subsp. *massiliense*: Our Experince

**DOI:** 10.3390/pathogens11121399

**Published:** 2022-11-23

**Authors:** Carmen Rodríguez-Cerdeira, Rigoberto Hernández-Castro, Carlos Daniel Sánchez-Cárdenas, Roberto Arenas, Alejandro Meza-Robles, Sonia Toussaint-Caire, Carlos Atoche-Diéguez, Erick Martínez-Herrera

**Affiliations:** 1Efficiency, Quality, and Costs in Health Services Research Group (EFISALUD), Health Research Institute, SERGAS-UVIGO, 36213 Vigo, Spain; 2Dermatology Department, Hospital do Vithas, 36206 Vigo, Spain; 3Department of Health Sciences, University of Vigo, 36310 Vigo, Spain; 4European Women’s Dermatologic and Venereologic Society (EWDVS), 36700 Tui, Spain; 5Psychodermatology Task Force of the Ibero-Latin American College of Dermatology (CILAD), Buenos Aires C1093, Argentina; 6Departamento de Ecología de Agentes Patógenos, Hospital General Dr. Manuel Gea González, Ciudad de México 14080, Mexico; 7Dermatology Department, Centro Médico Nacional La Raza, Paseo de las Jacarandas S/N, La Raza, Azcapotzalco, Ciudad de México 02990, Mexico; 8Dermatology Department (Section Mycology), Manuel Gea González Hospital, Mexico City 14080, Mexico; 9Histopathology Department, Manuel Gea González Hospital, Mexico City 14080, Mexico; 10Centro Dermatológico Dr. Fernando Latapí, Mérida Yucatán 29000, Mexico; 11Sección de Estudios de Posgrado e Investigación, Escuela Superior de Medicina, Instituto Politécnico Nacional, Plan de San Luis y Díaz Mirón S/N, Col. Casco de Santo Tomas, Alcaldía Miguel Hidalgo, Ciudad de México 11340, Mexico

**Keywords:** *Mycobacterium abscessus* complex, *Mycobacterium abscessus* subsp. *massiliense*, non-tuberculous mycobacteria, mesotherapy, cutaneous infections, molecular identification

## Abstract

Background: Members of *Micobacterium. abscessus* complex comprises three subspecies (*M. abscessus* subsp. *Abscessus*, *M. abscessus* subsp. *Bolletii*, and *M. abscessus* subsp. *Massiliense*) and are a rapid-growing nontuberculous mycobacteria present in different aquatic habitats and soil. It often causes a wide spectrum of infections involving pulmonary infections, surgical wound infections, and infections related to mesotherapy, catheters, hemodialysis devices, endocarditis, and disseminated infections in immunocompromised individuals. Methods: In this article we comment on the most relevant aspects of nine patients with skin lesions caused by *M. abscessus* subsp. *massiliense* infection. Clinical characteristics, histopathology, and molecular identification were performed. Results: The patients in the clinical cases presented a history of trauma, tattoos, and physical therapy techniques. The most common treatments were minocycline and clindamycin, doxycycline, ceftriaxone, cephalexin, moxifloxacin, rifampicin, and trimethoprim-sulfamethoxazole. The evolution of the treated patients was acceptable, except for one patient, who showed a partial improvement. *M. massiliense* were identified in all clinical cases using a species-specific PCR. Conclusion: Our series consisted of nine cases of skin biopsies recorded in different years; for this reason, we do not have all the data necessary for a complete description, in particular in four cases, causing limitations in the manuscript, especially in the therapy used and the evolution of patients due to lack of follow-up.

## 1. Introduction

Microorganisms that form the *Mycobacterium abscessus* complex are considered saprophytes and can be found in soil, water, mud, organic matter, animal feed deposits, sediments, and vegetables, for which the environment is the main source of infection. The importance of studying these microorganisms lies in their ability to survive in the absence of nutrients, grow at a wide range of temperatures, form biofilms, and resist the action of chlorinated disinfectants and glutaraldehyde [1,2]. They have been implicated as causal agents of wound and soft tissue infections [3], especially in immunocompromised patients and after invasive procedures, such as acupuncture and surgery [4,5].

Members of *M. abscessus* complex comprises three subspecies: *M. abscessus* subsp. *abscessus*; *M. abscessus* subsp. *bolletii* and *M. abscessus* subsp. *massiliense*. Taxonomy classification has been controversial due to the close relationship between *M. abscessus* subsp. *bolletii* and *M. abscessus* subsp. *massiliense*. However, whole-genome sequencing based phylogenetic analysis confirmed their differentiation in three distinct subspecies [6]. This differentiation is important because the treatment varies depending on the causal agent.

Skin and soft tissue infection by *M. abscessus* subsp. *massiliense* is difficult to differentiate from a skin infection by other nontuberculous mycobacteria or *Nocardia* because the clinical presentation is similar (persistence of subcutaneous nodules, erythema, and infected wounds) [7,8]. 

Here, we report nine clinical cases of cutaneous infection due to *M. abscessus* subsp. *massiliense* in Mexican patients and one Spanish patient. Due to the history of the treatments to which the patients were subjected, the lack of response to usual antibacterials, and the heterogeneity of the clinical manifestations, a granulomatous reaction associated with a foreign body was intentionally sought. Clinical characteristics, histopathology, and molecular identification were performed.

## 2. Material and Methods

Nine cases with cutaneous mycobacteriosis infection were processed. For all patients, skin biopsies were obtained. The histopathology study including hematoxylin-eosin and Ziehl-Neelsen staining. The biopsy samples were also processed for bacterial molecular identification. Briefly, genomic DNA was isolated from paraffin-embedded tissue samples using a DNeasy blood and tissue kit (Qiagen, Valencia, CA, USA) according to the manufacturer’s instructions before preliminary removal of paraffin by extraction with the xylene protocol [9]. Molecular identification was achieved using 16S rRNA and *mass_3210* gene amplification. For multiplex polymerase chain reaction (PCR) amplification, a set of primers previously reported to identify all mycobacterial species (506 bp of 16S rRNA gene), 5′-GAGATACTCGAGTGGCGAAC-3′ and 5′-CAACGCGACAAACCACCTAC-3′, were used. For *M. abscessus* subsp. *abscessus* and *M. abscessus* subsp. *Massiliense* identification a set of primers of *mass_3210* gene were applied (5′-GCTTGTTCCCGGTGCCACAC-3′ and 5′-GGAGCGCGATGCGTCAGGAC-3′) to amplify a 310 bp for *M. abscessus* subsp. *abscessus* and 1145 bp for *M. abscessus* subsp. *massiliense* [9]. The PCR products were visualized in a 1.5% agarose gel stained with ethidium bromide. 

## 3. Results

Data were collected from nine patients and sociodemographic and clinical characteristics are described in Table 1. The mean age of patients was 35.5 ± 16.6 years, ranging from 18 months to 60 years. Most of our patients were men (66.5%). The time of evolution was 12.1 ± 21.3 months, with a range of 0.1 months to 60 months. Two patients were HIV-positive, and one received anti-TNF-α biological treatment. We emphasize that most patients had a history of trauma, tattoos, electromagnetic girdle, or anti-TNF-α biological drugs, while three patients did not have information on the origin of the infection. Following the morphological patterns, the most frequent presentations were erythema and infiltration (Figure 1 and Figure 2), followed by nodules and ulcers (Figure 3). Gum, shrink scars, and fistulas were observed (Figure 4). The histopathological studies showed that the most frequent lesion was suppurative and granulomatous dermatitis, followed by a fistulous tract and abscess. 

The molecular identification showed that the nine samples amplified a 506 bp product and a 1145 bp product, corresponding to *M*. *abscessus* subsp. *massiliense* (Figure 5).

With respect to the treatment in four patients, we did not have results on the treatment and the evolution, due to not being able to carry out the follow-up of the patients. The most common treatments were minocycline and clindamycin, followed by doxycycline, ceftriaxone, cephalexin, moxifloxacin, rifampicin, and trimethoprim-sulfamethoxazole. The evolution of the treated patients was generally acceptable, except for one patient, who showed only a partial improvement. 

## 4. Discussion

Mycobacterial species are ubiquitous in the environment in water and soil habitats and can be classified according to growth characteristics: the group of non-tuberculous mycobacteria belonging to fast-growing mycobacteria and the slow-growing mycobacteria. Currently, *M. abscessus* complex comprises three closely related subspecies: *M*. *abscessus* subsp. *abscessus*, *M*. *abscessus* subsp. *massiliense*, and *M*. *abscessus* subsp. *bolletii*. *M*. *abscessus* subsp. *abscessus* is more frequently associated with pulmonary infections; however, *M*. *abscessus* subsp. *massiliense* is more predominant in skin or underlying soft tissue infection [9,10,11].

During the last three decades, the frequency of diseases caused by the *M. abscessus* complex has increased by diverse factors: greater virulence of mycobacteria; increase in immunocompromised patients, especially patients with human immunodeficiency virus (HIV), patients with organ transplants, and widespread use of antineoplastic drugs in interventional therapies and cancer treatment; use of biological drugs for the treatment of inflammatory skin diseases, and progresses in methods or techniques to detect and identify these pathogen [12]. In our cases, we observed two immunosuppressed patients with HIV infection (Cases 1 and 7), an important comorbidity for *M*. *abscessus* subsp. *massiliense* infection.

Cases of infection by *M*. *abscesses* complex have been reported in patients with a history of laser myopia correction surgery, mesotherapy sessions, or breast implants [4,9,13]. Case 8 describes a patient with a history of mesotherapy, which is related to risk factors for developing the disease. The same occurs in patients with a history of surgical procedures, injections, minor trauma, and immunosuppressive treatment [3]. In cases 2, 3, 6 and 9, the patients had a history of minor trauma due to contact with thorns, tattoos, electromagnetic girdle, and acupuncture, respectively.

The clinical presentation of skin and soft tissue infections is variable and includes pyogenic subcutaneous abscesses with an acute inflammatory reaction, violaceous erythematous nodules, dermatitis, cellulitis, folliculitis, ulcers, and chronic inflammatory reactions with fistula formation. These infections are characterized by a late onset of symptoms between 2 and 14 weeks after the history of inoculation [12,14,15]. In our clinical cases, patients present skin manifestations characterized by the presence of nodules, ulcers, erythema and infiltration, gums, fistulas, and shrink scars and the most histopathological lesion was suppurative and granulomatous dermatitis. 

Regarding treatment, we must consider the extension, anatomical site, and the immunological status of the host. Due to the resistance observed in the treatments used, the use of combinations of these and extraction of foreign bodies or debridement of the lesions is recommended. An important characteristic of *M*. *abscessus* complex is the presence of macrolide resistance observed in *M*. *abscessus* subsp. *Abscessus*, and *M*. *abscessus* subsp. *bolletti* due to functional inducible erythromycin ribosome methyltransferase (*erm* (41)), generating a resistant phenotype. However, in *M*. *abscessus* subsp. *massiliense* the *erm* (41) gene is not functional, generating a profile susceptible to macrolides and useful as a treatment for this microorganism [8,16]. Tigecycline and moxifloxacin are also more effective against *M*. *abscessus* subsp. *massiliense* than against *M*. *abscessus* subsp. *abscessus* [10,17].

Some authors have reported successful treatment of *M. abscessus* complex with a combination use of macrolides with amikacin, fluoroquinolones, imipenem/cilastatin, or cefoxitin [18,19,20,21,22]. In addition, a few cases were successfully treated with oral clarithromycin as monotherapy [23,24]. In our patients, the treatments were similar, using the same type of antibiotics; however, in four clinical cases, the treatment and evolution data were not available (Table 1). The evolution of the treated patients was generally acceptable, except for one patient, who showed only a partial improvement. 

For *M*. *abscessus* subsp. *massiliense* identification, conventional culture methods for bacterial isolation needs at least 7 days of incubation and in many times presenting a low level of success. The use of biochemical tests is not useful because they are not able to differentiate the members of *M*. *abscessus* complex. The histopathologic findings of *M*. *abscessus* complex are not specific, which may represent a diagnostic problem to the unexperienced pathologist [11]. The use of molecular techniques allows a rapid and more accurate identification of *M*. *abscessus* complex and are a fundamental tool for the control of *Mycobacterium* infections. 

For exact identification of *M*. *abscessus* complex species, sequencing analysis of 16S rRNA subunit, the internal transcription spacer 16S-23S (ITS-1), and *hsp65* and *rpoB* genes are recommended [8]. To carry out the identification, we used a previously described PCR end point, designed from the whole-genome sequence analysis, which can differentiate the species of *M*. *abscessus* complex. Among these specific genes, *mass_3210* (879 bp) is specific to *M*. *abscessus* subsp. *massiliense* and is located between *mass_3209* and *mass_3211*, which correspond to *mab_3265*c and *mab_3266*c genes, respectively, in the *M*. *abscessus* subsp. *abscessus* genome. This insertion gene region can be used to differentiate *M*. *abscessus* subsp. *abscessus* and *M*. *abscessus* subsp. *bolletti* from *M*. *abscessus* subsp. *Massiliense* by designing primers to sites flanking *mass_3210*, to positions in *mab_3265*c/*mass_3209* and *mab_3266*c/*mass_3211* genes [12,25]. In this sense, the correct identification of species or subspecies in relatively short times will allow improved patient management and therapeutic treatments, as well as epidemiological actions. 

## 5. Conclusions

Our series consisted of nine cases of skin biopsies recorded in different years; for this reason we do not have all the data necessary for a complete description, in particular in four cases, causing several limitations in the work, especially in the therapy used and the evolution of patients due to lack of follow-up. We carried out the molecular identification of *M*. *abscessus* subsp. *massiliense* as the etiological agent of various skin infections using a PCR capable of differentiating *M*. *abscessus* complex members. The accurate identification of *M*. *abscessus* complex is important because antibiotic susceptibility and treatment outcomes differ according to the etiologic *M*. *abscessus* complex members. In addition, in the case of the Spanish patient, the finding of *M*. *abscessus* subsp. *massiliense* was associated with treatment with an anti-TNF drug, this is remarkable because we have not found any similar cases in the literature reviewed.

## Figures and Tables

**Figure 1 pathogens-11-01399-f001:**
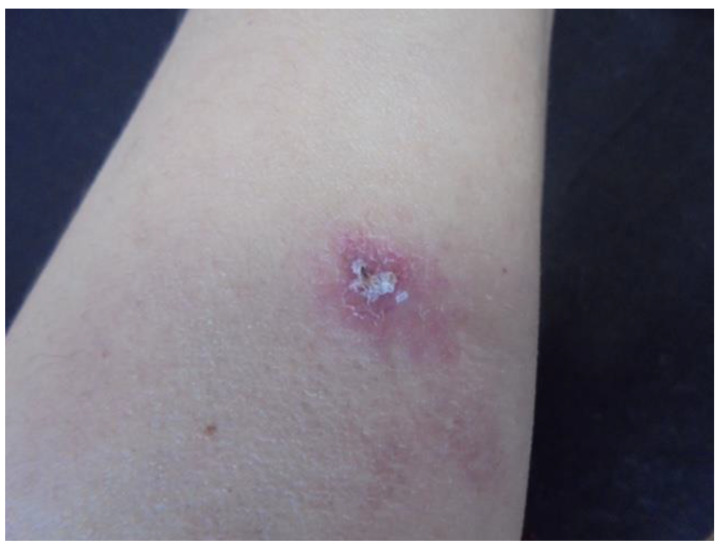
Erythema and infiltration in the left forearm (Patient 1).

**Figure 2 pathogens-11-01399-f002:**
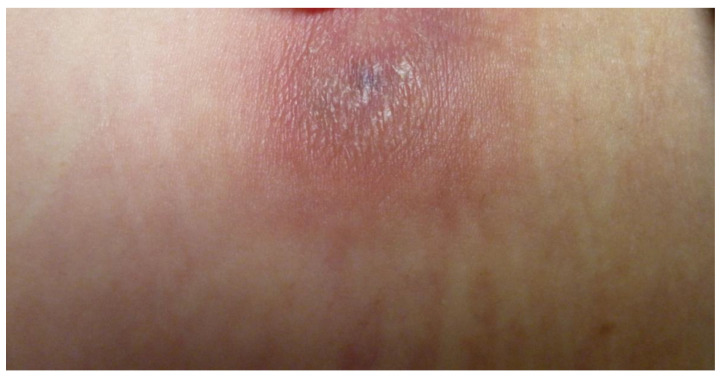
Infiltrated lesions in the abdomen (Patient 8).

**Figure 3 pathogens-11-01399-f003:**
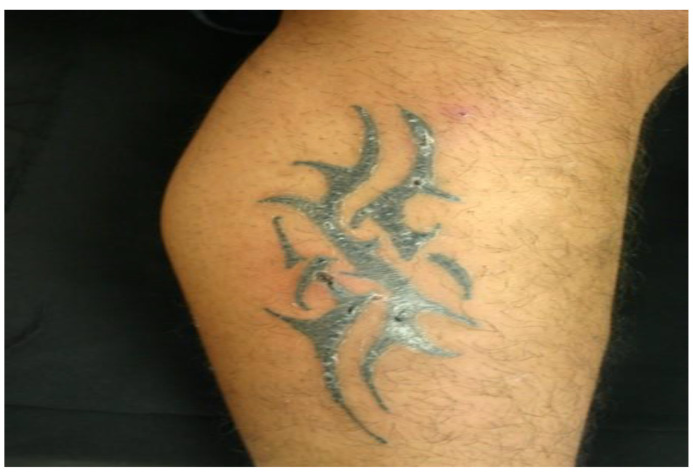
Nodular lesion on the right leg under a tattoo (Patient 3).

**Figure 4 pathogens-11-01399-f004:**
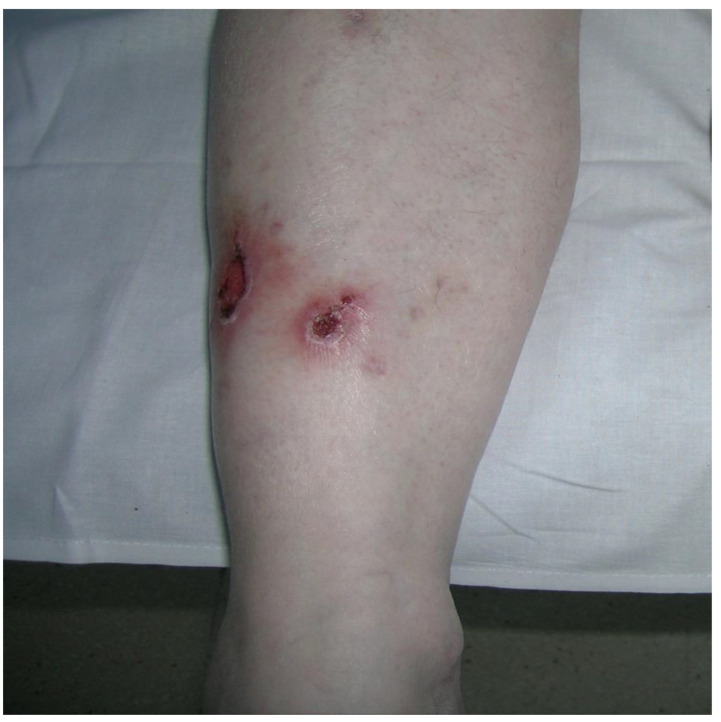
Erythema and shrink scars located throughout the lower of the right leg (Patient 9).

**Figure 5 pathogens-11-01399-f005:**
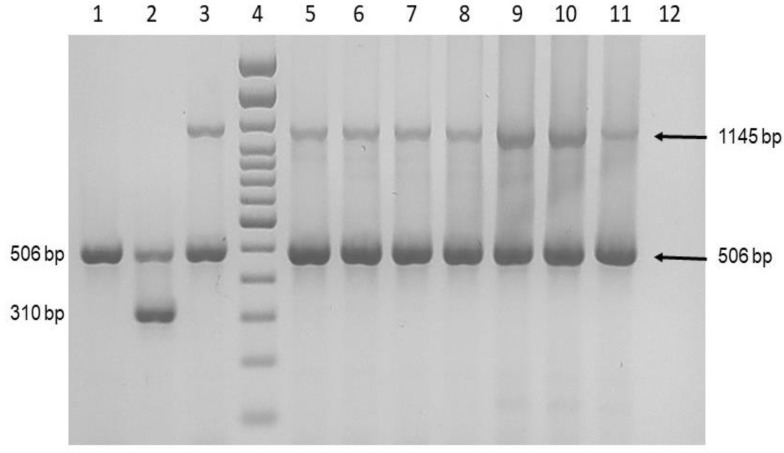
Molecular identification of *M. abscessus*. Multiplex PCR for the amplification of *Mycobacterium* spp., *M. abscessus* subsp. *abscessus*, and *M. abscessus* subsp. *masiliense*. Lane 1: *M. tuberculosis* 322; Lane 2: *M. abscessus* subsp. *abscessus* GEA-Ped1; Lane 3: *M. abscessus* subsp. *massiliense* GEA-Mic1; Lane 4; 100 bp DNA ladder; Lanes 5–11: Patient 1–7; Lane 12: Negative control.

**Table 1 pathogens-11-01399-t001:** Clinical and sociodemographic characteristics of the patients.

Cases	1	2	3	4	5	6	7	8	9
Sex	Male	Female	Male	Male	Female	Male	Male	Female	Male
Age (years)	29	25	18	24	55	23	28	56	60
Clinical manifestations	Nodules	Ulcers	Nodules	Nodules and ulcers	Gums	Shrink scares and fistulas	Erythema and infiltration	Erythema and infiltration	Erythema and infiltration
Location of lesions	Face, forearm and left leg	Perianal	Right thigh	Lower limb right	Upper limb right	Trunk	Both upper and lower limbs	Abdomen	Lower limb right
Evolution (months)	1	36	0.1	0.4	4	60	5	0.5	2
Comorbidities/ Personal history	HIV	Trauma	Tattoo	-	-	Electromagnetic girdle	HIV	-	Anti-TNF-α biological drug treatment
Histophatology	CD8+ lymphoproliferative disorder	Suppurative and granulomatous dermatitis	Suppurative and granulomatous dermatitis	Suppurative and granulomatous dermatitis	Suppurative and granulomatous dermatitis	Fistulous tract	Suppurative and granulomatous dermatitis	Abscess	Suppurative and granulomatous dermatitis
PCR	*M*. *abscessus* subsp. *massiliense*	*M*. *abscessus* subsp. *massiliense*	*M*. *abscessus* subsp. *massiliense*	*M*. *abscessus* subsp. *massiliense*	*M*. *abscessus* subsp. *massiliense*	*M*. *abscessus* subsp. *massiliense*	*M*. *abscessus* subsp. *massiliense*	*M*. *abscessus* subsp. *massiliense*	*M*. *abscessus* subsp. *massiliense*
Treatment	-	-	Doxycycline 200 mg/orally, every day for 6 weeks; Minocycline 200 mg/orally, for 6 weeks; Ceftriaxone 2 g/IM/daily /14 days Rifampicin 300 mg orally/daily/ 12 weeks	-	Trimethoprim/sulfamethoxazole 60 mg/800 mg /12 h Minocycline 200 mg/orally, for 6 weeks	Cephalexin 500 mg and 300 mg clindamycin every 8 h, both orally and for 6 weeks	-	Clindamycin 300 mg, orally, every 8 h for 6 weeks	Moxifloxacin 400 mg, orally for 8 weeks

- = NA = not applicable or not available.

## Data Availability

Not applicable.

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
