# Peer review of "Atypical Mycobacteriosis Due to Mycobacterium abscessus subsp. massiliense: Our Experince"

_pathogens, 2022, doi:10.3390/pathogens11121399_

Round 1
Reviewer 1 Report
In this study, the author reported atypical mycobacteriosis due to Mycobacterium abscessus subsp. massiliense from nine patients. My main concern is the identification of massiliense. M. abscessus subsp. massiliense is 1 of 3 subspecies of M. abscessus. M. abscessus subsp. massiliense has an identical 16S rRNA gene sequence to the other 2 subspecies, Mycobacterium abscessus subsp. bolletii and Mycobacterium abscessus subsp. abscessus, but can be differentiated by rpoβ and erm41 gene sequencing. Here, molecular identification was performed using 16S rRNA and mass3210 gene amplification by multiplex PCR. They did not do sequencing to confirm the PCR products considering the whole tissue genomic DNA template and non-specific amplification. So mass3210, rpoβ, and erm41 gene sequencing is required to confirm massiliense infection. Or isolation and identification of massiliense from the tissue would be confirmative. Some minors: 1) format of abstract. 2) Why is it atypical mycobacteriosis due to massiliense infection compared with others? It would be helpful if the authors highlight this background in Introduction. 3) Hematoxylin-eosin and BAAR staining were performed according to the statement in this study. What these data say? It’s better to have these data and analysis. 4) The significance of this observation of atypical mycobacteriosis should be discussed in abstract and discussion.
Author Response
Reviewer 1
First of all, thank you very much for the time spent reviewing our paper. We will then send you the answers.
In this study, the author reported atypical mycobacteriosis due to Mycobacterium abscessus subsp. massiliense from nine patients. My main concern is the identification of massiliense. M. abscessus subsp. massiliense is 1 of 3 subspecies of M. abscessus. M. abscessus subsp. massiliense has an identical 16S rRNA gene sequence to the other 2 subspecies, Mycobacterium abscessus subsp. bolletii and Mycobacterium abscessus subsp. abscessus, but can be differentiated by rpoβ and erm41 gene sequencing. Here, molecular identification was performed using 16S rRNA and mass3210 gene amplification by multiplex PCR. They did not do sequencing to confirm the PCR products considering the whole tissue genomic DNA template and non-specific amplification. So mass3210, rpoβ, and erm41 gene sequencing is required to confirm massiliense infection. Or isolation and identification of massiliense from the tissue would be confirmative.
Multiplex PCR design and interpretation.
For Mycobacterium species exclusion we have used the protocol described in 2017 by Chae y cols.:
Chae H, Han SJ, Kim S-Y, Ki C-S, Huh HJ, Yong D, Koh W-J, Shin SJ. 2017. Development of a one-step multiplex PCR assay for differential detection of major Mycobacterium species. J Clin Microbiol 55:2736 –2751.
https://doi.org/10.1128/ JCM.00549-17.
Note that the subspecies bollletii has not been excluded against because it has not been historically present in our environment.
Assay procedure:
The assay utilizes an eight-target multiplex PCR. The sizes of the resulting PCR products and the groups that they identify are as follows: a 506-bp amplicon specific to the 16S rRNA gene for all mycobacteria. After that we AMPLIFY a 310-bp amplicon and a 1,145-bp amplicon specific to mass_3210 for M. abscessus and M. massiliense, respectively.
As table 1: Second PCR expected products
mass_3210 gene 5’ GCTTGTTCCCGGTGCCACAC 3’ M. abscessus lenght: 310 nt
5’ GGAGCGCGATGCGTCAGGAC 3’ M. massiliense Lengh: 1145 nt
Some minors: 1) format of abstract.
We did
2) Why is it atypical mycobacteriosis due to massiliense infection compared with others? It would be helpful if the authors highlight this background in Introduction.
In the cases that we report, due to the history of the treatments to which the patients were subjected, the lack of response to usual antibacterials, and the heterogeneity of the clinical manifestations, a granulomatous reaction associated with a foreign body was intentionally sought, and was able to identifyby PCR multiplex to M. abscessus subsp. massiliense M. Thanks to this, therapy could be directed towards the etiological agent with good results in most patients.
3) Hematoxylin-eosin and BAAR staining were performed according to the statement in this study. What these data say? It’s better to have these data and analysis.
BAAR stain, we mean in this section to Zielh Nielsen stain. We have already corrected it
The Histopathological studies showed that the most frequent lesion was suppurative and granulomatous dermatitis, followed by fistulous tract and abscess.
Results of the Histopathological studies are collected in table 1
Ziehl-Neelsen stained biopsies did not show acid-fast mycobacteria.
4) The significance of this observation of atypical mycobacteriosis should be discussed in abstract and discussion.
We did
Reviewer 2 Report
Major point
1) There are already reports about nodule infection of M. abscessus, similar to your research. What is the difference and novelty of this study between these studies?
Minor points
1) There are some errors in spelling and citation. Authors should correct these errors and recheck the manuscript more carefully.
2) In the Material and Methods section, What/How is BAAR staining? It is not familiar to me, so please add the brief method and its corresponding.
3) You should revise the sentence from lines 151 to 158.
In line 152, the sentence is “… increased owning to three fundamental factors”
But from lines 151 to 158, there are four topics from a) to d) as fundamental factors for M. abscessus complex.
4) You should simplify the discussion section because that is including those not related to this study.
Author Response
First of all, thank you very much for the time spent reviewing our paper. We will then send you the answers.
Reviewer 2
25 Oct 2022 13:43:45
Final del formulario
© 1996-2022 MDPI (Basel, Switzerland) unless otherwise stated
Major point
- There are already reports about nodule infection of abscessus, similar to your research. What is the difference and novelty of this study between these studies?
The disease associated with non-tuberculous mycobacteria is the most common type of infection associated with mesotherapy, and at this time in Latin America any of the aesthetic techniques is being seen. In our cases, these etiologies did not occur, since only one patient had a history of tattoos. Taking into account that infections by microorganisms that form the Mycobacterium abscessus complex can be confused with other pathologies, with suppurative and non-suppurative granulomatous inflammation. Clinicians must be alert and think that any scenario for this germ may be possible
In the case of the patient, the finding of M. abscessus subsp. massiliense from Spanish was associated with treatment with an anti-TNF drug, this is remarkable because we have not found any similar case in the literature reviewed.
Minor points
- There are some errors in spelling and citation. Authors should correct these errors and recheck the manuscript more carefully.
We did
2) In the Material and Methods section, What/How is BAAR staining? It is not familiar to me, so please add the brief method and its corresponding.
BAAR stain, we mean in this section to Zielh Nielsen stain. It was a error.
For its stain we need to follow the next steps:
Place the fixed slides on the staining rack according to their order number, spreading side up. The blades should be separated by an interval of 1 cm and never touch each other.
Cover the slides one after the other with Ziehl's 0.3% carbolic fuchsin working solution, filtered
By placing a strip of absorbent paper such as filter paper or even newspaper, the staining solution will be retained and the deposit of fuchsin crystals on the smear will be avoided.
Heat the slides from below using the flame of a Bunsen burner, alcohol lamp or cotton ball soaked in alcohol, until steam is emitted. The dye solution should never be boiled. Do not allow the dye to dry out
Leave the slides covered with a hot and steaming solution of carbolic fuchsin for 5 minutes, ironing the flame if necessary
Rinse the slides gently with water to remove excess carbol fuchsin
Drain the excess rinsing water from the slides. Sputum smears are red in color.
Then
Cover the slides with 25% sulfuric acid or an alcohol acid solution and leave to act for 3 minutes, after that the red coloration should have almost disappeared. If necessary, repeat this sequence for two more minutes .
Gently wash off the sulfuric acid or alcohol-acid and excess food coloring with water. Drain excess rinse water from the slides.
Finally
Cover the slides one after the other with the counterstaining solution (0.3% methylene blue) and leave to act for 1 minute
Rinse the slides with water individually
Drain the water from the slides and let them air dry
3) You should revise the sentence from lines 151 to 158.
In line 152, the sentence is “… increased owning to three fundamental factors”
But from lines 151 to 158, there are four topics from a) to d) as fundamental factors for M. abscessus complex.
We have corrected the text and also those sentences
4) You should simplify the discussion section because that is including those not related to this study.
We did
Round 2
Reviewer 1 Report
All my comments have been addressed.
Reviewer 2 Report
This manuscript has been revised well.
I think this paper can be acceptable.